# Evaluation of All-for-One Tourism in Mountain Areas Using Multi-Source Data

**Hou Jiang [1,2], Yaping Yang [1,3],\* and Yongqing Bai [1,2]** 

1   State Key Laboratory of Resources and Environmental Information System, Institute of Geographic Sciences and Natural Resources Research, Chinese Academy of Sciences, Beijing 100101, China; jiangh.18b@igsnrr.ac.cn (H.J.); baiyq@lreis.ac.cn (Y.B.)
2   College of Resources and Environment, University of Chinese Academy of Sciences, Beijing 100049, China
3   Jiangsu Center for Collaborative Innovation in Geographical Information Resource Development and Application, Nanjing 210023, China
\*   Correspondence: yangyp@igsnrr.ac.cn; Tel.: +86-10-6488-9045

**Abstract:** All-for-one tourism is a new viewpoint of tourism development involving overall planning and cooperative mechanisms. Over the past few years, the researchers have put forward many conceptual models to guide the top-level design and specific practice of all-for-one tourism. However, these studies mainly focus on social, economic and cultural effect in mature tourism areas, lacking comprehensive analysis from geographical perspective and neglecting the underdeveloped regions. In this paper, we attempt to apply geographic information system technology to tourism evaluation, exploring the approach of all-for-one tourism development in mountain regions. Zunyi city is selected as the research region and evaluated on the abundance, quality and spatial pattern of tourism resources, climate comfort, natural disaster possibility, and convenience of infrastructure or social service. Multi-source datasets collected from websites, reanalysis data, remote sensing products and observation stations are used. Based on data analysis, some recommendations including enriching cultural tourism products through cultural creativity, ensuring regional coordinated development through spatial optimization, respecting the spatiotemporal characteristics of climate and the laws of nature, and strengthening construction of infrastructure, are discussed to promote the healthy development of all-for-one tourism.

**Keywords:** all-for-one tourism; spatial analysis; tourism evaluation; mountain areas

## 1. Introduction

Tourism, an international industry and the biggest provider of jobs, plays a vital role in promoting national or regional economic growth [1]. Since the period of reform and opening up, the tourism industry has been flourishing in China, and large numbers of attractions, scenic spots, restaurants and hotels have been developed for international or domestic tourists. According to the statistics provided by Ministry of Culture and Tourism of the People's Republic of China, the annual comprehensive contribution of the national tourism industry is about 9.13 trillion yuan, contributing to 11.04% of the total GDP, and tourism provides employment opportunities for 108.15 million people directly or indirectly, accounting for 10.28% of the total employment-population in 2017. However, the traditional scenic spot tourism has sunk into difficulty in driving the development of the regional economy over the past decades [2,3]. Thus, all-for-one tourism is put forward to adapt to the trend of new changes and promote the transformation and upgrading of tourism [4,5]. It is a new tourism mode aiming at developing a project in partnership with all stakeholders and involving in overall planning and cooperative development of all industries [6–12]. In recent years, all-for-one tourism has attracted the

attention of governments, enterprises and social problem researchers, as it is regarded as an effective and reliable approach to achieve rural revitalization and promote the coordinated and sustainable development of urban and rural areas [6].

Recently, various conceptual models have been presented to explain the connotation of all-for-one tourism. Li et al. [7] summed up all-for-one tourism as "four new ideas" (resources, products, industries and markets) and "eight constructive aspects" (all tourism elements, all professions, all processes, all levels, the entire space and time, the whole society, all participant departments and all latent tourists). They argued that all-for-one tourism should aim at providing tourists with sufficient product experiences by integrating various industries, cooperating different departments, uniting all participants in the region, and taking full use of tourism destinations. Li [8] pointed out that all-for-one tourism is a new regional coordinated development mode, where tourism is the dominant industry and promotes the development of the economy and society in the chosen region. The ultimate goal of all-for-one tourism is to achieve the scientific and reasonable integration of resources, industries and developments, as well as social co-construction and sharing through optimizing and upgrading the complex systematic structure within a specific region. Zhang et al. [9] emphasized the domain perspective of all-for-one tourism, including temporal-spatial domain, industry domain, element domain and management domain. Feng [5] unscrambled the necessity of all-for-one tourism development from the five characteristics of tourist destination summarized by Cooper et al. [10], and proposed to grasp development law, strengthen the problem-oriented principle and consider comprehensive impacts in the process of promoting transformation from scenic spots to all-for-one tourism. Wang [11] holds the view that all-for-one tourism is used to promote the innovation of ideas on tourism development and to push forward the reform of the administrative system of tourism in China. It advocates the idea of the sharing economy, developing a project in partnership with all stakeholders and protecting the freedom of individual participation in social life. These studies explored the connotation and development routes of all-for-one tourism, and describe the top-level design and developing blueprint of all-for-one tourism either in large natural regions or in small towns. However, these studies mainly focus on social, economic and cultural effects in mature tourism areas, while comprehensive evaluations from a geographic perspective are scarce.

For a long time, most attention has been paid to scenic spot tourism in the tourism community. For example, Potschin [12] analyzed how the quality of site-level environmental assessment could be improved by using the concept of natural capital. Yi and Hu [13] analyzed the relationship between tourist attractions and all-for-one tourism and put forward four development strategies of tourist attractions under the background of all-for-one tourism. Zhang [14] analyzed the internal factors affecting the development of the tourist attractions through a strengths weaknesses opportunities and threats (SWOT) analysis, qualitative analysis and quantitative analysis on China's tourism resources. Their research datasets are from in-situ survey, site observation and field sampling, usually lacking objectivity, comprehensiveness or temporal-spatial continuity, thus it is difficult to meet the demand of overall planning and decision-making for all-for-one tourism. With the arrival of the era of big data, tourism research has brought forth amazing improvements. Zhu et al. [15] established a tourism resources attraction evaluation system using collected relevant information from the internet and analyzed the spatial pattern of tourism resources' attraction in Beijing. Yang et al. [16] advocated that the large scale of big data could finely make up for the limitation of sample size issues faced by survey data users, providing a new way to understand tourist behavior. Similarly, Li et al. [17] argued that big data analytics could provide sufficient data without sampling bias, helping both academia and industries to better understand tourist behavior. Xiang et al. [18] claimed that big data analytics could develop new knowledge to reshape the understanding of hospitality industry and to support the corresponding decision-making. In all, big data allows a better understanding of tourism demand, tourist behavior, tourist satisfaction and other tourism issues.

In this paper a mountain tourism city, Zunyi, is chosen as the research region for a case study aiming to evaluate the suitability and explore the approach for all-for-one tourism development in mountain

areas from a geographic perspective. The temporal-spatial analyses are based on multi-source datasets collected from websites, reanalysis data, remote sensing products and observation stations. Many specific factors related to development of tourism industry are taken into consideration in the process of temporal-spatial analysis, including abundance, quality and spatial distribution of tourism resource, local topography, temperature, precipitation, relative humidity, surface wind speed, sunshine hour per day, land cover types, traffic facilities, hotels etc. On the basis of data analysis, some recommendations are proposed for development of all-for-one tourism, such as improving cultural tourism products through cultural creativity, ensuring regional coordinated development through spatial optimization, respecting the spatiotemporal characteristics of climate and the laws of nature, and strengthening the construction of infrastructure.

## 2. Data and Methods

### 2.1. Research Region

Zunyi City covers an area of 30.7 thousand km$^2$, locating in the northeast of Guizhou province, China (Figure 1). It possess abundant natural tourism resources, such as Chinese Jurassic Park, Liyuan Grassland Tourism Resort, Shuanghe Karst Cave National Geological Park, etc. At the same time, Zunyi is the national historical and cultural city in China, with world cultural heritage Hailongtun, world natural heritage Chishui Danxia. In addition, the subtropical monsoon climate is suitable for outdoor mountain tourism. It is cool and humid all year round, with abundant rainfall and sunshine, and there is no severe cold in winter and intense heat in summer [19,20]. In addition, Zunyi is a sacred place for red tourism in China [21], known as "the turning city, the capital of conference". In 1935, the Communist Party of China held the famous Zunyi Conference, which became a turning point of the party's life and death. In 2016, Zunyi was regarded as the national demonstration area of mountain all-for-one tourism by the Ministry of Culture and Tourism of the People's Republic of China. Therefore, Zunyi is selected as the research area to explore the approach to developing all-for-one tourism in mountain areas from the geographical aspects in this paper.

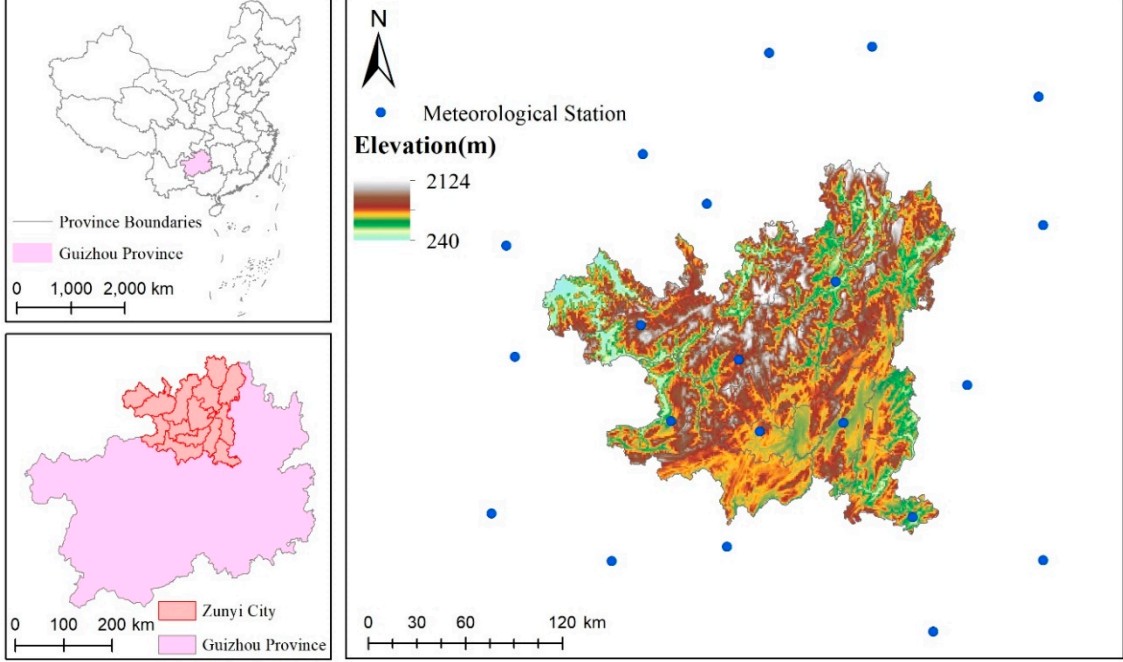

**Figure 1.** Location and DEM of Zunyi City. The meteorological stations are used for downscaling other raster datasets described in Section 2.3.

*2.2. Basic Data*

From a collection of relevant literature, discussions with experts, professional research reports and on-site surveys, we organized the overall architecture for evaluation of all-for-one tourism as shown in Figure 2. Three main aspects are taken into consideration, i.e., tourism resources, natural conditions and social service. Specifically, we attempt to estimate the abundance, quality and spatial characteristics of tourism resources. Climate comfort reflects a tourist's feelings under different meteorological conditions and is of great importance to the tourism industry, especially vacation tours [22]. It is influenced by many factors, such as air temperature, relative humidity, surface wind speed, and solar radiation etc., and there exists complex relationships among different factors. Thus, thermal environment model and cold environment model are used for comprehensive evaluation of natural conditions in the following sections. Meanwhile, natural disaster risk is another important factor and it might relate to altitude, slope, precipitation, surface vegetation, etc. Similarly, various elements are involved in social service, such as transportation, accommodation, delicacies, medical treatment, communication, customs etc. To conduct the temporal-spatial continuous evaluation, multi-source datasets are utilized. The specific datasets are listed as follows:

- Digital Elevation Model (DEM). DEM data used in this study is SRTM data (Shuttle Radar Topography Mission, NASA-NGA, USA). SRTM successfully generates the most complete high-resolution digital topographic database of the Earth, covering over 80% of the Earth's land surface between 60° north and 56° south latitude with data points posted every 1 arc-second (approximately 30 m). The elevation models are arranged into tiles, each covering one degree of latitude and one degree of longitude. The raw data can be obtained from the website: http://srtm.csi.cgiar.org/SELECTION/inputCoord.asp. Figure 1 shows the elevation of Zunyi, which ranges from 240 to 2124 m.

- NCEP/NCAR Global Reanalysis Products. It is a continually updated globally gridded data set that represents the state of the Earth's atmosphere, incorporating observations and numerical weather prediction (NWP) model output from 1948 to present. It is a joint product from the National Centers for Environmental Prediction (NCEP) and the National Center for Atmospheric Research (NCAR). Its monthly mean datasets are used in our study, including monthly mean air temperature, monthly mean of relative humidity and monthly surface wind speed. Their spatial resolution is about 2.5°. The data can be obtained at: ftp://ftp.cdc.noaa.gov/pub/Datasets/ncep.reanalysis.derived/surface/.

- Geospatial Data Cloud. It provides MODIS synthetic products in China. The datasets used in this paper includes China 1 KM land surface temperature (LST) monthly synthetic products (MODLT1M) and China 500 M normalized difference vegetation index (NDVI) monthly synthetic products (MODND1M). MODLT1M is the monthly average value of MODLT1T daily products while MODND1M is the maximum value of MODND1D daily products within a month. The data set is provided by International Scientific & Technical Data Mirror Site, Computer Network Information Center, Chinese Academy of Sciences. Registered users can freely obtain data from the website: http://www.gscloud.cn.

- SPOT-NDVI Data. The SPOT VGT-S NDVI data at 1 km spatial resolution can be obtained from http://www.spot-vegetation.com/. The VGT-S NDVI product is the synthesized NDVI for a 10-day period. The 10-day periods were defined as the 1st to 10th, the 11th to the 20th and the 21st to the end of each month. Thus, there are three NDVI images for each month. The monthly composite DNVI images were derived from corresponding three 10-day NDVI products through calculating their maximum value. The VGT-S product has been pre-processed for geometric, radiometric and atmospheric corrections. Therefore, no further pre-processing of the SPOT VGT-S product is needed for practical application.

- TRMM 3B43 Data. The Tropical Rainfall Measuring Mission (TRMM) was launched in November 1997 as a joint project by NASA and the Japanese Space Agency (JAXA). The mission uses five

instruments: Precipitation Radar (PR), TRMM Microwave Imager (TMI), Visible Infrared Scanner (VIRS), Clouds & Earths Radiant Energy System (CERES) and Lightning Imaging Sensor (LSI). The TRMM Multi-satellite Precipitation Analysis (TMPA) was designed to combine all available precipitation datasets from different satellite sensors and monthly surface rain gauge data to provide estimation of precipitation at spatial resolution of 0.25°. One of TMPA products is the TRMM 3B43 monthly data, covering 50° N to 50° S for 1998–present. The latest Version 7 was available to public in May 2012 and can be obtained freely from http://mirador.gsfc.nasa.gov.

- In-situ Ground Meteorological Data. US National Climate Data Center is the world's largest provider of meteorological and climatic data. Its land-based observations are collected from instruments sited at locations on every continent. They include temperature, dew point, relative humidity, precipitation, wind speed and direction, visibility, atmospheric pressure, and types of weather occurrences. Data on sub-hourly, hourly, daily, monthly, annual, and multiyear timescales are available. All used stations are displayed in Figure 1. The related datasets are free on website: ftp://ftp.ncdc.noaa.gov/pub/data/gsod.

- POI Data. A point of interest (POI) is a specific location that someone may find useful or interesting. In this study, related POIs include unitary tourism resource points, traffic facilities, hotels or homestays, hospitals or casualty stations, restaurants or farmsteads. These POIs are very important for assessment of tourism resources and social service, and for understanding tourist behavior and satisfaction. Many map service companies, such as Google (https://cloud.google.com/maps-platform/), Baidu (http://lbsyun.baidu.com/), Amap (https://lbs.amap.com/) etc., support location-based services, and POIs can be obtained from their map open platforms through keyword retrieval.

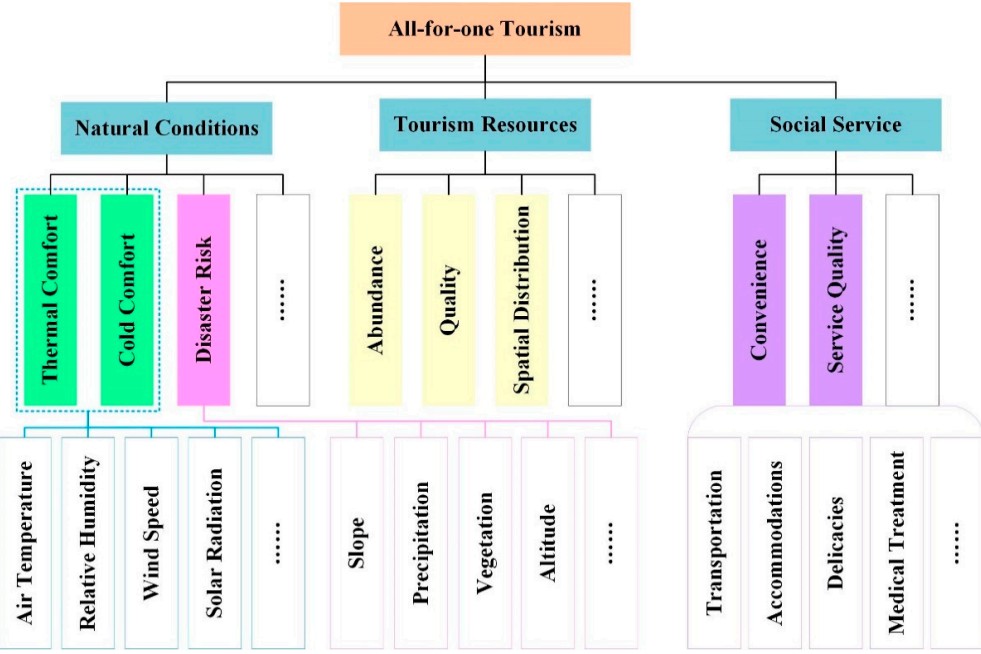

**Figure 2.** The overall architecture for assessment of all-for-one tourism development.

## 2.3. Downscaling Method

Over the past decades, development of satellite sensors has resulted in multiple sources of meteorological datasets, such as precipitation [23,24], land surface temperature [25,26], relative humidity [27,28] etc., that provide more reliable estimations over un-gauged areas compared with other interpolation methods. However, their spatial resolutions (i.e., 0.25–0.5°) are still too coarse for hydrological simulation and environmental modeling when applied to local basins and regions. Great

efforts have been made to advance the spatial downscaling algorithms of satellite-based meteorological datasets. For example, Immerzeel et al. [29] proposed an algorithm for downscaling TRMM-based annual precipitation datasets from 0.25° to 1 km by using the exponential function between the precipitation and NDVI. Jing et al. [30] implemented random forests (RF) and support vector machine (SVM) separately to downscale the yearly TRMM 3B43 V7 precipitation data from 25 km to 1 km over the Tibetan Plateau area based on precipitation-land surface characteristics.

In our study, the method in Immerzeel et al. [29] is used to downscale TRMM 3B43 data from 0.25° to 1 km, and the result is taken as inputs of the SVM-based method [30] to downscale monthly mean air temperature (AT), monthly mean of relative humidity (RH) and monthly surface wind speed (SWS) of NCEP/NCAR Global Reanalysis Products. Downscaled data will be further used for evaluation of natural conditions. DEM and NDVI are used as variables for air temperature downscaling. DEM, NDVI, land surface temperature and average precipitation are considered as independent variables for relative humidity and wind speed. Before downscaling, an additional process [31,32] is introduced to correct residuals of original reanalysis products using in-situ ground meteorological data. The used meteorological stations are shown in Figure 1. Figure 3 illustrates the flowchart for downscaling NCEP/NCAR products. More details are described in references [30–32].

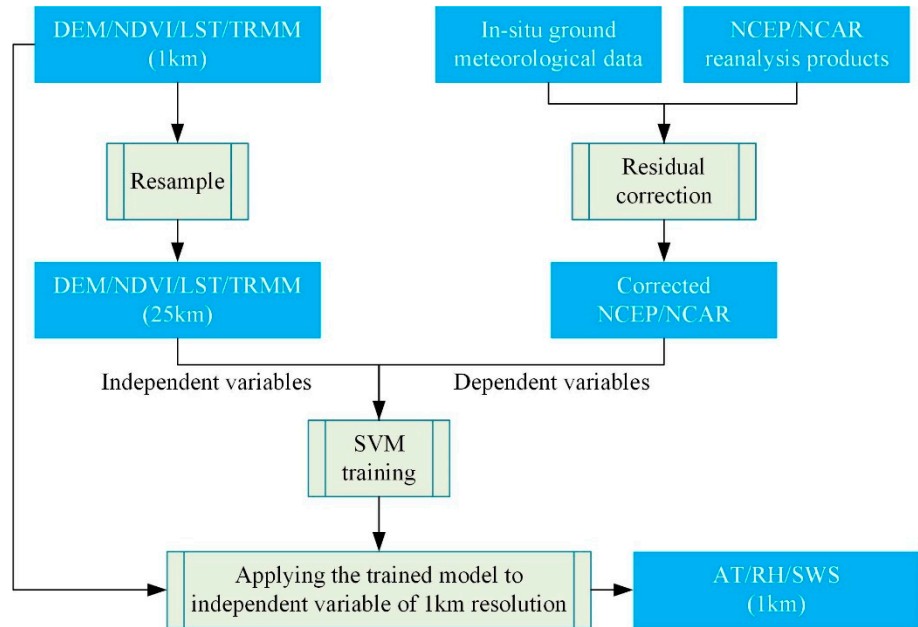

**Figure 3.** The flowchart of downscaling algorithm used in our study.

## 3. Results and Discussion

### 3.1. Tourism Resources

#### 3.1.1. Abundance and Quality

In 2003, the Ministry of Culture and Tourism of the People's Republic of China promulgated *Classification, Investigation and Evaluation of Tourism Resources (GB/T 18972-2017)* [33] as the national standard. However, due to the rise of leisure vacation and the richness of tourism products, the traditional methods have been unable to meet the needs for classification of diverse new types of tourism resources, especially urban tourism resources. Based on the analysis of the concept, elements and characteristics of various tourism resources, and combined with other research [34,35], we make some improvements and the final classification system is shown as Table 1. Furthermore, we invited 15 experts with tourism and geography background to assess the quality of 100 randomly sampled tourism resource units. This was a subjective process in which each expert evaluated the

general quality of tourism resource units based upon their judgement and knowledge on tourism. The questionnaires (Appendix B in [33]) containing basic descriptions on natural characteristics, infrastructure and developing status, together with on-site pictures, are sent to the experts who are required to score to these units following the evaluation system in Table 2. The system limits the maximum score of each evaluation factor and the scoring range of their corresponding four-levels, representing "excellent", "perfect", "good" and "poor", respectively. Another added value can be optionally given to the unit concerning its situation on environmental protection or safety, i.e., negative 5 points for seriously polluted status, negative 4 points for moderately polluted status, negative 3 points for slightly polluted status, while positive 3 points for status existing environmental protection measures.

**Table 1.** Types of tourism resources in Zunyi City. No. = number, P = proportion of total.

| Main Category | | Subcategory | | | Basic Type |
|---|---|---|---|---|---|
| | | No. | P (%) | Subcategory Name (No. of Basic Types) | Types |
| Natural tourism resources | A. Geographical Landscape | 142 | 17.66 | AA. Comprehensive natural tourist destination (50) AB. Sedimentary and tectonic (3) AC. Geomorphic processes (79) AD. Natural Changes (10) | 24/37 |
| | B. Water Landscape | 67 | 8.33 | BA. Rivers (18) BB. Natural wetlands and marshes (30) BC. Waterfalls (13) BD. Springs (6) | 9/15 |
| | C. Ecological Landscape | 39 | 4.85 | CA. Woodland (17) CC. Flowered areas (14) CD. Wildlife habitat (8) | 8/11 |
| | D. Astronomical and Climate Landscape | 11 | 1.37 | DA. Light phenomena (4) DB. Weather and climate phenomena (7) | 5/8 |
| | Subtotal | 259 | 32.21 | 13 | 46/71 |
| Manmade and cultural tourism resources | E. Heritage sites | 148 | 18.41 | EA. Prehistoric site (6) EB. Cultural and economic monument (142) | 7/12 |
| | F. Architecture and infrastructure | 367 | 45.65 | FA. Comprehensive cultural tourism destinations (97) FB. Dedicated activity sites (36) FC. Landscape architecture and attached buildings (160) FD. Residential areas and communities (50) FE. Burial grounds (16) FF. Transportation architecture (3) FG. Hydraulic structures (5) | 35/49 |
| | G. Tourism products | 17 | 2.11 | GA. Local tourism products (17) | 2/7 |
| | H. Cultural Activities | 13 | 1.62 | HA. Memorials (2) HB. Art (1) HC. Folk customs (5) HD. Modern festivals (5) | 6/16 |
| | Subtotal | 545 | 67.79 | 14 | 50/84 |
| | Grand total | 804 | 100 | 26/31 (83.87%) | 96/155 (61.94%) |

**Table 2.** Evaluation system of tourism resources.

| Evaluation Item | Evaluation Factor | Score | Levels |
|---|---|---|---|
| Resource element value | Sightseeing & recreation value | 30 | 30-22; 21-13; 12-6; 5-1 |
| | Historical, cultural, science and Artistic value | 25 | 25-20; 19-13; 12-6; 5-1 |
| | Rare & singularity | 15 | 15-13; 12-9; 8-4; 3-1 |
| | Scale, abundance and probability | 10 | 10-8; 7-5; 4-3; 2-1 |
| | Integrity | 5 | 5-4; 3; 2; 1 |
| | Visibility & influence | 10 | 10-8; 7-5; 4-3; 2-1 |
| Resource influence | Appropriate tour period or using range | 5 | 5-4; 3; 2; 1 |
| Added value | Environmental protection & environmental safety | 4/+ | -5; -4; -3; 3 |

Statistical results show that there are 804 tourism resource units, covering 8 main categories, 26 out of 31 (83.87%) of subcategories and 95 out of 155 (61.94%) of basic types. It reveals that Zunyi boasts of varied landscapes and a rich combination of tourism resource types. The final average score of expert evaluation is 87.74 out of 100. According to these scores, tourism resources are divided into three grades according the standard [33] (Section 6.3.2.2). The units whose average score are larger than 90 are referred to as "excellent tourism resources"; if the average score is between 75 and 90, the unit is referred to as "perfect tourism resources"; while the others are treated as "general tourism resources". In total, 33 out of 100 evaluated tourism resources are excellent, 28 are perfect and 39 are general. Although the personal perception of experts might affect the objectivity of the final results, this generally shows that tourism resources in Zunyi are of high quality. It is evident that abundant categories and excellent quality of tourism resources have laid a solid foundation for tourism development in Zunyi.

However, it is notable that although the main categories A-F account for more than half of the basic types, the proportion of tourism products (G) and cultural activities (H) are lower than 50% (only 2/7 and 6/16, respectively). In addition, the excellent units are mainly from natural tourism resources, while most manmade resources are of low-quality, especially category E and F. Therefore, it is particularly important to improve the quality of manmade tourism resources and enrich cultural tourism products through cultural creativity. Under the traditional tourism mode, tourists just visit the scenic spots and take pictures to record beautiful scenes or wonderful moments. It is difficult to satisfy tourists' multi-level and diversified modern requirements for beautiful natural scenes, unique local cultural experiences, or age-old minority amorous feelings [36]. Thus, all-for-one tourism should pay attention to the creation of innovative cultural tourism products and the interaction between tourists and local culture. Firstly, traditional cultural resources can be directly utilized as tourism products. We can use traditional villages, cultural relics, museums, memorials, art galleries, world cultural heritage, intangible cultural heritage exhibition halls and other cultural sites to carry out cultural experience activities. Secondly, existing tourism products can be upgraded through cultural integration, focusing on in-depth excavation of historical culture, regional characteristic culture, national folk culture, traditional farming culture, and traditional handicraft culture. Thirdly, tourism can be combined with other industries. For example, in the field of agriculture, it is strongly advocated to develop sightseeing agriculture, leisure agriculture, creative agriculture such as pastoral landscape or balcony agronomy, and other agricultural patterns with tourism function such as customized agriculture, exhibition agriculture, family farming or family pasture.

3.1.2. Spatial Characteristics

Tourism resources are the foundation of the tourism industry. Different from fragmented scenic spots, all-for-one tourism treats a whole region as a tourist destination with complete functions to satisfy various tourists to achieve the integration of inside and outside scenic spots. The aim is to make everyone become a tourist and everywhere become tourism landscapes. The ideal state is that all tourism resources are evenly distributed spatially. Thus, it is vital to get knowledge of the spatial distribution of all tourism resources. Figure 4a shows the spatial distribution of natural and cultural tourism resources. Overall, different types are cross-distributed spatially, blending together. It is helpful to provide tourists with diverse sightseeing experiences, containing not only the relaxation and beauty of nature, but also solemnity or joyfulness from colorful historical or national culture.

However, the regional distribution of tourism resources is uneven, i.e., abundant in the central and western regions while few in the eastern region, as shown in Figure 4b. We further analyze the spatial density of tourism resources at pixel level. The kernel density analysis is widely used in the analysis of spatial characteristics and spatial patterns of geographical features [37], which is helpful to judge regional imbalance. It calculates the density of features in a neighborhood around those features.

Let $(x_1, x_2, \ldots, x_n)$ be a univariate independent and identically distributed sample drawn from some distribution with an unknown density *f*. Its kernel density at point *x* is:

$$\hat{f}_h(x) = \frac{1}{nh}\sum_{i=1}^{n}K(\frac{x - x_i}{h}), h > 0 \tag{1}$$

where *K* is the kernel, a non-negative function that integrates to one; *h* represents the bandwidth, which should be finely selected in practice. In our study, *h* is set to be the default value defined by the algorithm in ArcGIS [38]. Figure 4c shows the result of kernel density analysis. Multiple clustered centers are formed in space, especially in the northwestern and southwestern. The result is in agreement with results present in Figure 4b.

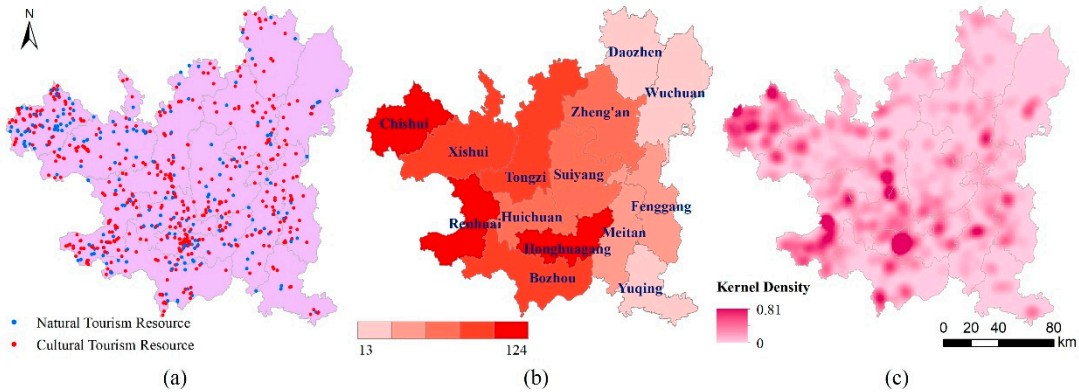

**Figure 4.** Spatial analysis of tourism resources. (**a**) Spatial distribution; (**b**) Regional statistics; (**c**) Kernel density.

The uneven spatial distribution of tourism resources remains a large obstacle to all-for-one tourism development. The government planning departments and tourism developers should pay more attention to discover and mine tourism landscapes in areas where tourism resources are scarce. Under the premise of ecological protection, natural landscapes must be developed appropriately. A more advisable approach is to integrate tourism industry with regional development. For example, in the process of rural revitalization, local residents can vigorously develop agricultural sightseeing, leisure tour, pastoral experience, or hold folk culture festivals, food tasting and feasting activities etc. Meanwhile, it must be emphasized that the development of all-for-one tourism must adjust to local conditions and highlight local characteristics. Every district should explore their unique approach through innovation to avoid having sluggish tourism attractions and products. The charm of all-for-one tourism lies in letting a hundred flowers bloom and being rich and colorful. For example, Danxia landform in Chishui and Xishui, red culture in Honghuagang and Bozhou, Gelao culture in Daozhen and Wuchuan, wine culture in Renhuai, and tea culture in Meitan can be the typical characteristics.

Currently, not all districts or towns in Zunyi meets the conditions to develop all-for-one tourism. Local tourism development committee should focus on constructing tourism infrastructure, creating top tourism products, widening the coverage of the tourism industry and improving the influence of tourism brands. To promote the development of all-for-one tourism, it is necessary to pay attention to the social and cultural impact of the tourism industry on tourism destinations and to take measures to promote the understanding of different cultures, protect the traditional cultural heritage, improve people's leisure level and life quality and eliminate the negative impact imposed by tourism development. It is also needed to attach great importance to the impact of tourism on the coordinated development of urban and rural areas. In the process of blending tourism into traditional rural and agricultural development, some problems and shortcomings, such as the guarantee of farmers' interests after excessive capitalization or semi-urbanization, should be taken seriously.

Another way to eliminate unbalanced development is to strengthen regional cooperation. The idea of tourism development must be transformed from the traditional scene sightseeing to all-for-one tourism mode. All-for-one tourism emphasizes regional comprehensive development concept, which requires breaking through the simplicity and closeness of scenic spots and extending the radiative effects of central tourism areas. It is necessary to strengthen cooperation with adjacent regions and jointly develop typical tourist routes or areas by linking various tourist attractions. Relying on the special geographical location, Zunyi can combine the tourism resources of Chongqing, Sichuan and Guizhou to design top-quality tourism routes, such as the red tourism line of the Long March, ecological tour zone of the Golden Triangle, and ethnic cultural tourism areas.

### 3.2. Natural Conditions

### 3.2.1. Climate Comfort

Climate is a salient resource for tourism and a dominant attribute of a tourist destination [22,39]. It has a major effect on tourism demand, satisfaction and decision-making [40] since tourists are sensitive to climate and climate change [41]. Thus, it is vital to assess the suitability of climate for tourism for the sake of decision-making by tourism participants. For instance, tourism planners could better evaluate a destination for tourism development and incorporate climate in infrastructure planning and programming; the insurance industry might design diverse weather insurance products for the tourism industry; tourists can choose a destination and take out insurance on likelihood of poor weather conditions occurring while on holidays. Researchers have made considerable efforts to devise climate indices owing to the multifaceted nature of weather and the complex ways the weather variables come together to give meaning to climate for tourism. In this study, temperature-humidity index (THI) [42] and wind effect index (WEI) [43] are chosen to assess the thermal comfort and cold comfort, separately.

Thermal comfort is the condition of mind that expresses human physiological satisfaction under the influence of temperature and humidity. THI is calculated by mean of dry bulb temperature and relative humidity and its expression is as followed [42]:

$$THI = t_d - 0.55(1 - 0.01RH)(t_d - 14.5) \tag{2}$$

where $t_d$ represents the dry bulb temperature (°C), $RH$ is relative humidity (%). Herein, monthly mean air temperature and relative humidity of NCEP/NCAR reanalysis products preprocessed through method in Section 2.3 are used for calculation of THI. The values of 2013 locate in the range [9.2, 28.5] and are reclassified into five categories: Cold, cool, comfortable, warm and hot (Table 3) according to Kyle's study of the bioclimatic environment [44].

**Table 3.** Classification of temperature-humidity index (THI) and wind effect index (WEI).

| THI (°C) | Category | WEI |
|---|---|---|
| −1.7~13.0 | Cold | −800~−600 |
| 13.0~14.0 | Cool | −600~−300 |
| 15.0~20.0 | Comfortable | −300~−200 |
| 20.0~26.5 | Warm | −200~−50 |
| 26.5~29.9 | Hot | −50~80 |

Cold comfort evaluates the suitability for outdoor activities by taking into consideration the effects of surface wind speed and solar radiation [45]. The expression of WEI is [43]:

$$WEI = -(10\sqrt{V} + 10.45 - V)(33 - T) + 8.55S \tag{3}$$

where *V* means surface wind speed (m/s), *T* is air temperature (°C), and *S* is sunshine hour per day (h/d). Herein, monthly mean air temperature and surface wind speed of NCEP/NCAR reanalysis products preprocessed through method in Section 2.3 are used for calculation of WEI. Spatial continuous sunshine hours are obtained by means of interpolating in-situ measured data through Kriging method. The WEI values of 2013 vary in the range [−742, 74] and are divided into five levels (Table 3) referring to Terjung's research [43].

Figure 5 shows the variation of THI of Zunyi in different seasons. It is clear that spring (April 2013) is the best, followed by autumn (October 2013). Although summer (July 2013) is overheating in local, most of the regions belong to warm. Winter (January 2014) tends to be cold and humid, with obvious boundary along the mountain from northeast to southwest. Such characteristics can mainly be attributed to the joint influence of topography and monsoon. In summer, Zunyi is affected by the southwest monsoon from the Indian Ocean, thus low-latitude areas in the southwest are obviously hotter than others, while high-altitude areas remain warm environments because of the vertical zonal effect. In winter, northwest monsoon dominates, but is greatly weakened by the Tibetan Plateau. Therefore, the northwest is much colder, but would not reach the freezing level. In contrast, the southeast is relatively warm as the northwest mountains further weaken the cold wind. In addition, Figure 5c shows the average THI from 2003 to 2013, indicating that the climate in Zunyi tends to be cool and comfortable. Figure 6 shows the spatial distribution of WEI in different seasons. We can draw a similar conclusion to THI. In contrast, a larger region tends to be comfortable and only high-altitude mountains above about 2000 m reach a cold level. Nevertheless, the average level of WEI from 2003 to 2013 belongs to the cool type in the entire space from the perspective of cold comfort. All in all, warm and comfortable environments create excellent conditions for mountain travel or outdoor hiking, attracting urban residents who are eager to return to nature and enjoy themselves.

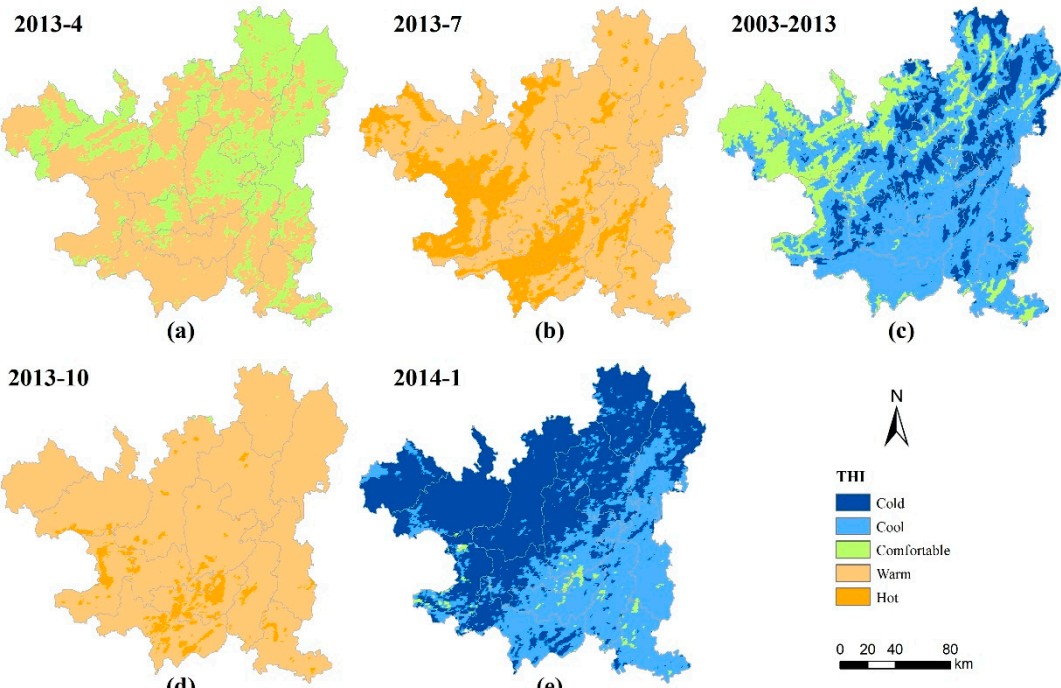

**Figure 5.** Variation of temperature-humidity index in different seasons. (**a**) Spring (April, 2013); (**b**) Summer (July, 2013); (**c**) Multi-year average (2003–2013); (**d**) Autumn (October, 2013); (**e**) Winter (January, 2014).

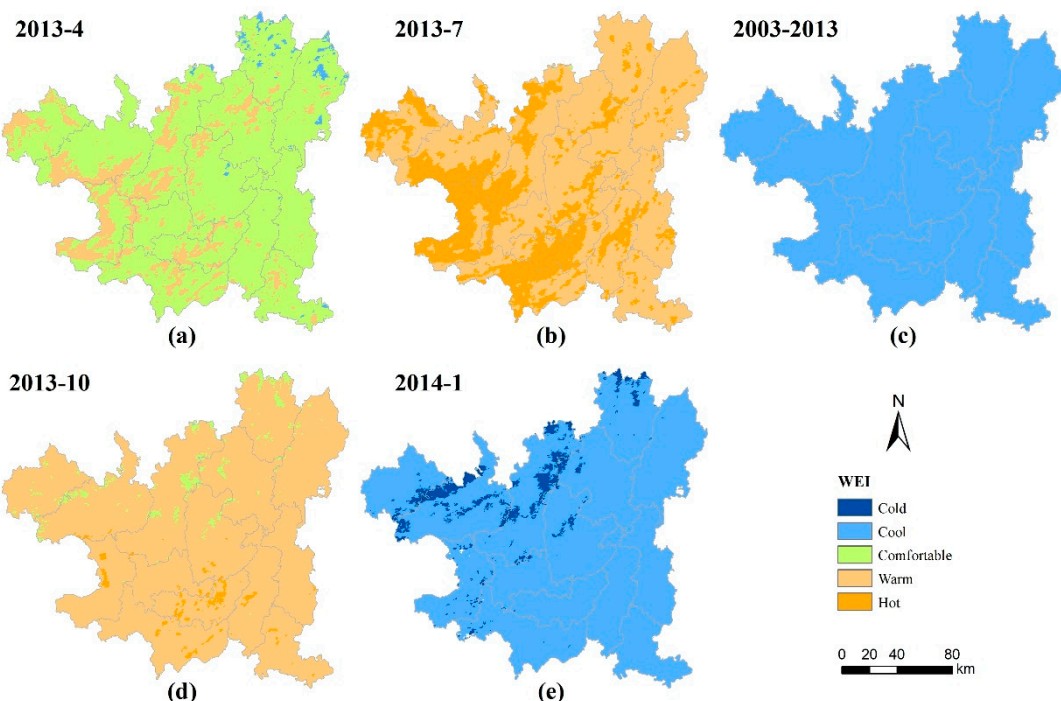

**Figure 6.** Variation of wind effect index in different seasons. (**a**) Spring (April, 2013); (**b**) Summer (July, 2013); (**c**) Multi-year average (2003−2013); (**d**) Autumn (October, 2013); (**e**) Winter (January, 2014).

The characteristics of the climate in Zunyi can be described as "four distinctive seasons, comfortable spring, warm autumn, no severe cold in winter, and no intense heat in summer". It means that Zunyi is suitable for developing tourism all the year round, without an obvious peak season and off season, but with different scenery in different seasons. Moreover, due to the complex topography and relative high altitude, the climate shows the characteristics of diversity and vertical zonality. Generally, the whole region can be roughly divided into four vertical climate zones: Mid-subtropical climate in hilly and valley areas, northern subtropical climate in low-altitude mountain areas, northern temperate climate in middle mountain areas, and the middle temperate climate in mountain areas above 1500 m of sea level. Different vertical climatic zones in mountain areas affect the distribution of temperature, moisture, soil, animals and plants, resulting in various and distinctive geographical, hydrographic, ecological, or climatic landscapes [46]. Tourism planners should grasp the distribution of tourism resources and master the spatial and temporal characteristics of different landscapes so that they are able to develop high-quality tourist attractions and design reasonable tourist sightseeing routes adapting to seasonal changes and spatial variation to avoid singularity and repetition of tourism products. Zunyi is strongly recommended to focus on "summer leisure tourism" comparing to the torrid climate in most areas of China during summer. According to the statistics provided by China Tourism Academy, the number of domestic tourists for summer vacation (July and August) reached 1 billion in 2017 in China, accounting for 20% of the total number of tourists in the year. In addition, global warming will positively affect mountainous plateau areas or middle- and high-latitude zones, attracting more tourists for summer vacation thanks to their relatively cool environments [47,48].

3.2.2. Evaluation of Disaster Risk

Mountains provide a thoroughly challenging environment for special sports and leisure activities which attract aficionados such as mountaineers, paragliders, or downhill skiers [49]. Mountain tourism, especially mountain adventure tourism, differs from general mass tourism in lowland regions due to its higher requirements on the safety of the tourism area [50]. It is highly necessary to evaluate disaster risks for mountain tourism. Zunyi stands in the transition zones from the Yunnan-Guizhou Plateau to the Hunan hills and Sichuan basins with undulating topography and complex landforms.

The area of mountains in Zunyi reaches 65.08%, the hilly area accounts for 28.35%, while the flat dam and valley basin is only 6.57%. Meanwhile, rainfall is abundant in Zunyi. In summer, as southwest monsoon moves northward, the water vapor from the Bay of Bengal and the Indian Ocean increases dramatically, and Zunyi enters plum rains season with heavy and frequent rainfall. The statistics from Ministry of Natural Resources show that Zunyi is faced with a high possibility of geologic hazards, such as landsides, collapse, and mudslides.

This section intends to gain spatial continuous results of disaster possibility in Zunyi referring to the method in ref. [51]. Four inducing factors are taken into consideration: Slope (Figure 7a), precipitation (Figure 7b), NDVI (Figure 7c) and altitude (Figure 1). Note that the precipitation is a 10-year average value from 2005–2015 and NDVI is 1-year average value in 2015. Each factor is divided into five levels (1, 2, 3, 4, 5) through the Jenks natural breaks classification method [52,53] and the higher level means a greater possibility of disaster risk. Then the weighted sum of all factors is used for evaluation of comprehensive disaster possibility. Herein, the weights are 0.574, 0.291, 0.090, and 0.045 for slope, precipitation, NDVI and altitude, respectively. All weights are determined by analytic hierarchy process (AHP) [54], a subjective method that can reflect difference of relative importance among indexes properly. Figure 7d shows the final assessment results. Let the possibility 100% correspond to the maximum value 5, then the possibility 90%, 80%, 70%, 60% equals 4.5, 4, 3.5, and 3, respectively. In the northeast and the central mountain area, the disaster possibility is over 90% due to large slope and excessive precipitation. In contrast, the possibility in the southwest is relatively small with a large slope but little precipitation and dense vegetation. Other regions have a low possibility, mainly thanks to the flat terrain.

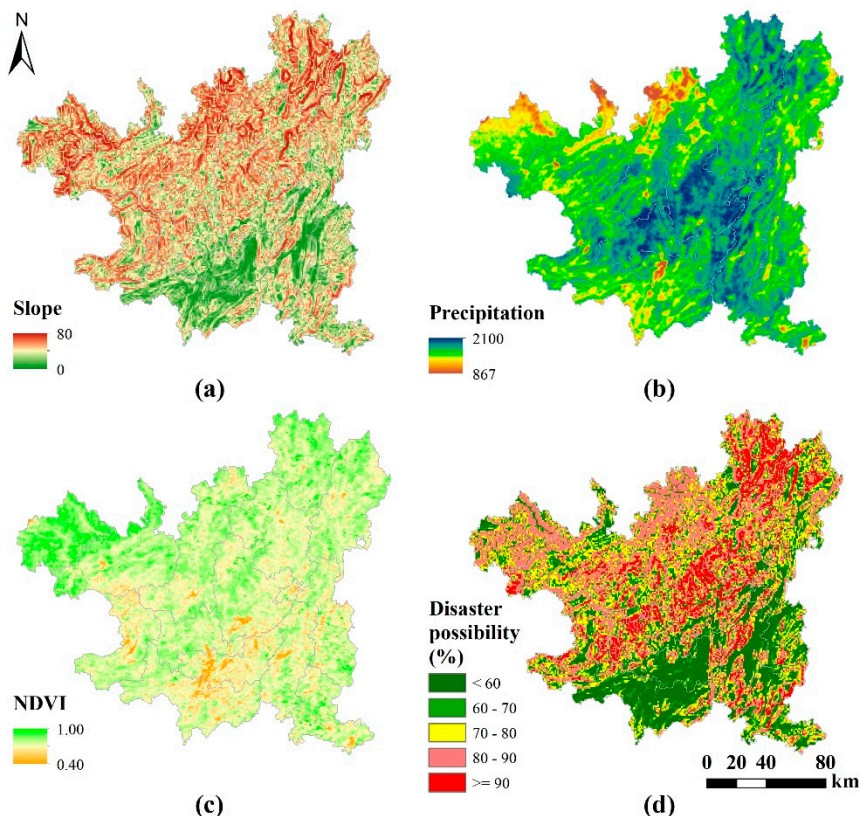

**Figure 7.** Evaluation of natural disaster risk. (**a**) Slope; (**b**) Precipitation; (**c**) NDVI; (**d**) Results of disaster risk. Note that the precipitation is 10-year average from 2005–2015 and NDVI is 1-year average in 2015.

During the development of tourism, humans should respect the laws of nature and avoid exploitation in areas with a fragile natural ecology and high risk of natural disasters. Promoting

all-for-one tourism does not mean developing tourism everywhere. Tourism development must be compatible with the carrying capacity of resources and the environment. If damage from environmental pollution caused by development of the tourism industry exceeds an acceptable ecological security threshold, it will threaten ecological service functions and the sustainable development of tourism areas [55]. Unfortunately, the diverse stakeholders often violate the objective laws for their immediate interests and contrasting goals during the process of tourism development. Innovative approaches are required to aid in the conflict resolution process. Research has shown that as the level of agreement between groups of stakeholders increases, so does the likelihood of collaboration and compromise [56]. At the local level, communities represented by local government officials are usually responsible for coordinating the relationships between public authorities, business owners, and other social stakeholders. In addition, collaboration with the academic community needs to be enhanced. Professional planning measures are needed to realize rational layout and optimal allocation of resources, facilities, elements, functions and industries in an all-round way so as to better relieve and reduce the pressure on core resources and ecological environments. From the perspective of humanistic concern, tourist-warning signs should be set up in particular locations to remind tourists to stay away from danger. At the same time, administrative departments should establish early-warning system to monitor the main inducing factors and assess the risk of natural disaster in real time, thus proper preventions can be arranged in advance to minimize the economic loss and casualties.

### 3.3. Social Service

Satisfying tourists is the final goal of tourism service as well as an important guarantee for the long-term development of tourism industry. Tourist satisfaction is a comprehensive judgment and psychological evaluation on tourism landscapes, natural and cultural environments, infrastructure, and social services etc. Through map open platforms, we have accumulated a large number of POIs relating to tourism, which are classified into four categories: Traffic facilities, hotels or homestays, hospitals or casualty stations, restaurants or farmsteads. Note that traffic facilities include passenger stations, railway stations, airports, entrance to expressway or main road. In this section, we measure the convenience of every spatial point by calculating its nearest Euclidean distance to POIs, considering the difficulty of constructing topological relations of spatial paths. The weighted average distance of all types of POI is treated as the final convenience of social services. All weights are determined through AHP and the values are 0.537, 0.232, 0.139, and 0.092 for traffic, hotel, hospital and restaurant, respectively.

Figure 8a shows the direct results of weighted average distance, which is divided into five levels (1, 2, 3, 4, 5) through the Jenks natural breaks classification method as shown in Figure 8b. Furthermore, the grading results are overlapped with results of land cover classification, containing urban construction areas and rural residential lands interpreted from Landsat 8 OLI images in 2015. Figure 8b shows that almost all urban areas belong to level 5, areas with the highest convenience of social services, while most rural areas are located in level 4 and 3, and extremely few people live in level 2 and 1. It indicates that the urban areas occupy most of the social resources while infrastructure in rural areas is relatively backwards.

Although this article only analyzes the limited resources, the problems reflected are universal. With the development of urban and rural economy and the improvement of farmers' living standards, the demand for public services is increasing. Currently, one of the most urgent need for the rural revitalization is the rural infrastructure construction related to the improvement of people's livelihood, including road construction, water supply, medical treatment, garbage collection, toilet improvement, street lighting, public transport etc. All-for-one tourism is an important approach to realize the strategy of rural revitalization. In order to revitalize the tourism industry, many initiatives need to be implemented simultaneously. On the one hand, great importance should be attached to rural infrastructure construction from the policy and investment. On the other hand, improvements are

highly required to perfect the legislation on the tracking, management, implementation, maintenance, investigation and evaluation of relevant rural funds and policies.

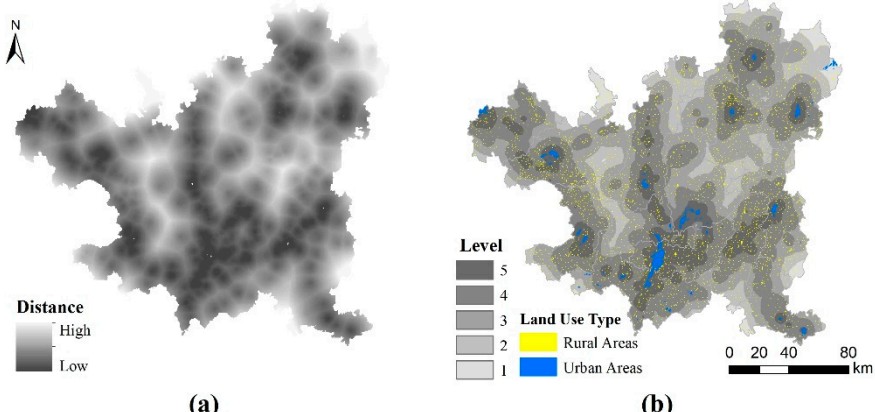

**Figure 8.** Evaluation of social service. (**a**) The results of weighted average distance; (**b**) The grading results of social convenience. Land cover types are interpreted from Landsat 8 OLI images in 2015.

The development of all-for-one tourism requires more convenient and high-quality transportation services. Transportation serves as a bridge to link tourists and tourism destination, especially with the rise of self-driving travelling the relationship between traffic and tourism becomes much closer. The local government must increase direct investment and guide enterprises' own investment to build up a complete tourism traffic network and improve public transport service facilities. Standardized traffic signs with distinctive visual effects and effective guidance should be set up on highways and main tour routes. Meanwhile, it is necessary to integrate traffic services with related tourism elements, such as tourism consultation, theme accommodation, the flavor of food and beverages, special shopping, and leisure and entertainment options. Parking lots, new energy vehicle charging piles, tourist toilets and other facilities are required to meet the traveling needs of tourists. It is also suggested to build an official tourism website, mobile phone app and public micro-signal in A-level scenic spots to provide tourists with a tour guide, route navigation and shopping services.

## 4. Summary

In this paper, we systematically analyzed the present situation of all-for-one tourism in Zunyi concerning tourism resources, natural conditions and social conditions from the geographical perspective. Some conclusions are drawn: (1) Zunyi possesses various high-quality landscapes and a rich combination of tourism resource types, but is relatively weak in providing cultural tourism products; (2) Different types of tourism resources are highly integrated, but the spatial distribution is uneven, i.e., abundant in the central and western while few in the eastern; (3) The climate is suitable for mountain tourism all year round and its variation in temporal and spatial domain breeds abundant natural tourism resources; (4) Some areas with large slope and excessive precipitation are in face of high risk of natural disaster; (5) The imbalance between urban areas and rural areas is a main obstacle to the development of all-for-one tourism. In fact, other regions in China are facing similar problems during the development of all-for-one tourism [15,20,34–37,40].

Based on the case study, some general recommendations are present to address the current problems for the healthy development of all-for-one tourism: (1) improving the quality of manmade tourism resources and enriching cultural tourism products through cultural creativity; (2) integrating tourism industry with regional development, strengthening regional cooperation with nearby tourism areas and focusing on construction of tourism infrastructure in backward areas; (3) designing tourist sightseeing routes adapting to the spatial and temporal characteristics of different landscapes; (4) respecting the laws of nature and avoiding exploitation in areas with fragile natural ecology

and high risk of natural disasters; (5) strengthening construction of infrastructure and building up a complete tourism traffic network. All-for-one tourism is a new viewpoint of tourism development involving in overall planning and cooperative mechanism. In practice, every participant should be aware of the complexity and comprehensiveness of each problem. For example, we should not only focus on the quantity of tourism resources, but also pay attention to their quality and spatial layout. Apart from emphasizing the investment and construction of infrastructure, great importance must be attached to the spatial balance of social services and coordinated development between urban and rural areas.

All-for-one tourism is a highly comprehensive and recapitulative concept. Our study concentrates on revealing objective phenomena or situations in temporal and spatial domain. In order to reflect its development connotation in a more comprehensive way, it is better to evaluate and analyze all-for-one tourism from many other perspectives and dimensions, such as personnel training. Restricted by the cultural level, many tour guides cannot provide professional interpretation service to meet the differentiated expectations of tourists. It is necessary to strengthen the ability and quality of tourism practitioners and promote the transformation of tour guides from explanatory and reception services to cultural and professional services. Tourism schools must shoulder the responsibility of cultivating compound, modernized, and international tourism talents by means of strengthening effective links with tourism industry, enterprises and society, promoting cooperation with governments, enterprises, other schools and international organizations.

The limitation of our study is based on the fact that all findings rely on the specific case and thus cannot be generalized. These experiences should not be copied, but altered adaptively when applying to other application scenarios. In addition, more datasets should be integrated into the evaluation process, such as statistical data on the total volume, the market characteristics, the consumption situation and the market trends in the source of tourists. At present, tourism management departments are actively promoting the accumulation of tourism statistics, the construction of tourism big data, and the application of data science and technology in tourism industry. With the continuous improvement of tourism-related methods and technology, it is reasonable to believe that the evaluation of all-for-one tourism will be more systematic and complete.

**Author Contributions:** Conceptualization, H.J. and Y.B.; Data curation, Y.B.; Funding acquisition, Y.Y.; Investigation, Y.Y.; Methodology, H.J.; Validation, Y.Y. and Y.B.; Writing—original draft, H.J.; Writing—review & editing, H.J. and Y.B.

**Funding:** This research was funded by the "Strategic Priority Research Program (A)" of the Chinese Academy of Sciences (No. XDA19020304), the Multidisciplinary Joint Expedition for China-Mongolia-Russia Economic Corridor (No. 2017FY101300), the Branch Center Project of Geography, Resources and Ecology of Knowledge Center for Chinese Engineering Sciences and Technology (No. CKCEST-2017-1-8), the National Earth System Science Data Sharing Infrastructure (No. 2005DKA32300), the Construction Project of Ecological Risk Assessment and Basic Geographic Information Database of International Economic Corridor Across China, Mongolia and Russia (No. 131A11KYSB20160091), and the National Natural Science Foundation of China (No. 41631177).

**Conflicts of Interest:** The authors declare no conflict of interest.

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
