# Peer review of "Evaluation of All-for-One Tourism in Mountain Areas Using Multi-Source Data"

_sustainability, doi:10.3390/su10114065_

Round 1
Reviewer 1 Report
Thank you very for addressing every single comment made in my original feedback. It was easy to see what you improved. You also provided suitable rational. Thank you!
That said, could you in your introduction make sure that somewhere that you mention clearly that this paper is a case study. In your conclusion mention the fact that the limitation is based on the fact findings can't be generalised.
I understand your paper better. I think that it gives another perspective of destination management (stakeholders, etc), but it does not advance knowledge that much. Apart from the 'all for one tourism' concept, I did not learn anything new.
Author Response
Point 1: Thank you very for addressing every single comment made in my original feedback. It was easy to see what you improved. You also provided suitable rational. Thank you! That said, could you in your introduction make sure that somewhere that you mention clearly that this paper is a case study. In your conclusion mention the fact that the limitation is based on the fact findings can't be generalised.
Response 1: Thank you for your advice. In the introduction, we have made it clear that this paper is a case study. “In this paper a mountain tourism city, Zunyi, is chosen as the research region for a case study aiming to evaluate the suitability and explore the approach for all-for-one tourism development in mountain areas from a geographic perspective.” In the conclusion, we stress this point again. “The limitation of our study is based on the fact that all findings rely on the specific case thus cannot be generalized. These experiences should not be copied, but altered adaptively when applying to other application scenarios. In addition, more datasets should be integrated into the evaluation process,”.
Point 2: I understand your paper better. I think that it gives another perspective of destination management (stakeholders, etc), but it does not advance knowledge that much. Apart from the 'all for one tourism' concept, I did not learn anything new.
Response 2: Thank you very much. In China, the current studies mainly focus on social, economic and cultural effect in mature tourism areas, lacking comprehensive analysis from geographical perspective and neglecting the underdeveloped regions. I think as for academia and practitioners, attentions should not only be paid to analysis of tourism development in mature areas, but also exploring the approach for all-for-one tourism development in backward areas. In addition, all-for-one tourism is regarded as an effective and reliable approach to achieve rural revitalization and promote the coordinated and sustainable development of urban and rural areas, especially in the underdeveloped regions. It is of great importance to design the scientific developing routes for the healthy development of all-for-one tourism. Although this paper might not introduce anything new to tourism researchers, I think our research would provide scientific evidence for decision-making on all-for-one tourism in mountains.

Reviewer 2 Report
Thank you for the opportunity to review the revised version of this article. Even at the first version, I have appreciated the novelty and the original approach, but raised some issues and made some suggestions to improve the article. I now consider it was significantly improved and reviewers suggestions have been followed.
Author Response
Point 1: Thank you for the opportunity to review the revised version of this article. Even at the first version, I have appreciated the novelty and the original approach, but raised some issues and made some suggestions to improve the article. I now consider it was significantly improved and reviewer’s suggestions have been followed.
Response 1: Thank you for your advice. Your previous comments are all valuable and very helpful for improving our paper, as well as the important guiding significance to further research.

Reviewer 3 Report
Dear authors,
I revised the first and the second version of the manuscript and I red many improvements.
My most important observation regards the methodology and the replicability of the analysis system. The selection of expert is always lacking and it is not clear whether an adequate selection has been made. From this depend all other results. So I suggest to describe this part in depth and verify the literature. See for example:
Muxika, I., Borja, Á., & Juan Bald. (2007). Using historical data, expert judgement and multivariate analysis in assessing reference conditions and benthic ecological status, according to the European Water Framework Directive. Marine Pollution Bulletin, 55(1–6), 16–29. http://doi.org/10.1016/J.MARPOLBUL.2006.05.025
The paradox of expert judgment in rivers ecological monitoring. (2016). Journal of Environmental Management, 184, 609–616. http://doi.org/10.1016/J.JENVMAN.2016.10.004
An investigation of dependence in expert judgement studies with multiple experts. (2017). International Journal of Forecasting, 33(1), 325–336. http://doi.org/10.1016/J.IJFORECAST.2015.11.014
Unleashing expert judgment in assessment. (2017). Global Environmental Change, 44, 1–14. http://doi.org/10.1016/J.GLOENVCHA.2017.02.005
Furthermore, the sentence "Furthermore, we invited 15 experts with tourism and geography background to score 100 235 randomly sampled tourism resource units referring to the evaluation system in Table 2." is not a result but a method then needs to be moved and described in the correct section.
Good luck!!
Author Response
Point 1: I revised the first and the second version of the manuscript and I red many improvements. My most important observation regards the methodology and the replicability of the analysis system. The selection of expert is always lacking and it is not clear whether an adequate selection has been made. From this depend all other results. So I suggest to describe this part in depth and verify the literature. See for example:
Muxika, I., Borja, Á., & Juan Bald. (2007). Using historical data, expert judgement and multivariate analysis in assessing reference conditions and benthic ecological status, according to the European Water Framework Directive. Marine Pollution Bulletin, 55(1–6), 16–29. http://doi.org/10.1016/J.MARPOLBUL.2006.05.025
The paradox of expert judgment in rivers ecological monitoring. (2016). Journal of Environmental Management, 184, 609–616. http://doi.org/10.1016/J.JENVMAN.2016.10.004
An investigation of dependence in expert judgement studies with multiple experts. (2017). International Journal of Forecasting, 33(1), 325–336. http://doi.org/10.1016/J.IJFORECAST.2015.11.014
Unleashing expert judgment in assessment. (2017). Global Environmental Change, 44, 1–14. http://doi.org/10.1016/J.GLOENVCHA.2017.02.005
Response 1: Thank you very much. I have read related literatures and learnt a lot. In section 3.1.1, we have described the process of expert evaluation in detail as follows “In 2003, the Ministry of Culture and Tourism of the People’s Republic of China promulgated Classification, Investigation and Evaluation of Tourism Resources (GB/T 18972-2017) [34] as the national standard. However, due to the rise of leisure vacation and the richness of tourism products, the traditional methods have been unable to meet the needs for classification of diverse new types of tourism resources, especially urban tourism resources. Based on the analysis of the concept, elements and characteristics of various tourism resources, and combined with other researches [35,36], we make some improvements and the final classification system is as Table 1. Furthermore, we invited 15 experts with tourism and geography background to assess the quality of 100 randomly sampled tourism resource units. This was a subjective process in which each expert evaluated the general quality of tourism resource units based upon their judgement and knowledge on tourism. The questionnaires (Appendix B in [34]) containing basic descriptions on natural characteristics, infrastructures and developing status, together with on-site pictures, are sent to the experts who are required to score to these units following the evaluation system in Table 2. The system limits the maximum score of each evaluation factor and the scoring range of their corresponding four-levels, representing “excellent”, “perfect”, “good” and “poor”, respectively. Another added value can be optionally given to the unit concerning its situation on environmental protection or safety, i.e., negative 5 points for seriously polluted status, negative 4 points for moderately polluted status, negative 3 points for slightly polluted status, while positive 3 points for status existing environmental protection measures.”. The part about data analysis is as “Statistical results show that there are 804 tourism resource units, covering 8 main categories, 26 out of 31 (83.87%) of subcategories and 95 out of 155 (61.94%) of basic types. It reveals that Zunyi boasts of varied landscapes and a rich combination of tourism resource types. The final average score of expert evaluation is 87.74 out of 100. According to these scores, tourism resources are divided into three grades according the standard [34] (Section 6.3.2.2). The units whose average score are larger than 90 are referred to as “excellent tourism resources”; if average score is between 75 and 90, the unit is referred to as “perfect tourism resources”; while the others are treated as “general tourism resources”. In total, 33 out of 100 evaluated tourism resources are excellent, 28 are perfect and 39 are general. Although the personal perception of experts might affect the objectivity of final results, it generally reflects that tourism resources in Zunyi are of high quality. It is evident that abundant categories and excellent quality of tourism resources have laid a solid foundation for tourism development in Zunyi.”
Point 2: Furthermore, the sentence "Furthermore, we invited 15 experts with tourism and geography background to score 100 235 randomly sampled tourism resource units referring to the evaluation system in Table 2." is not a result but a method then needs to be moved and described in the correct section.
Response 2: Thank you for your advice. We have corrected it.
